# Improving English-to-Indian Language Neural Machine Translation Systems

**Akshara Kandimalla [1,†], Pintu Lohar [2,†], Souvik Kumar Maji [1,\*] and Andy Way [2,\*]**

1    School of Computing, Dublin City University, D09 E432 Dublin, Ireland; akshara.kandimalla2@mail.dcu.ie
2    ADAPT Centre, Dublin City University, D09 Y074 Dublin, Ireland; pintu.lohar@adaptcentre.ie
\*    Correspondence: kumar.maji2@mail.dcu.ie (S.K.M.); andy.way@adaptcentre.ie (A.W.)
†    These authors contributed equally to this work.

**Abstract:** Most Indian languages lack sufficient parallel data for Machine Translation (MT) training. In this study, we build English-to-Indian language Neural Machine Translation (NMT) systems using the state-of-the-art transformer architecture. In addition, we investigate the utility of back-translation and its effect on system performance. Our experimental evaluation reveals that the back-translation method helps to improve the BLEU scores for both English-to-Hindi and English-to-Bengali NMT systems. We also observe that back-translation is more useful in improving the quality of weaker baseline MT systems. In addition, we perform a manual evaluation of the translation outputs and observe that the BLEU metric cannot always analyse the MT quality as well as humans. Our analysis shows that MT outputs for the English–Bengali pair are actually better than that evaluated by BLEU metric.

**Keywords:** machine translation; back-translation; parallel data

## 1. Introduction

The Indian languages belong to two major language families of the South-Asian subcontinent: *Dravidian* and *Indo-Aryan*. Although there are high numbers of native speakers in most Indic languages, there are still not enough available language-processing resources. Most of the available resources are in English. NMT requires a substantially large, high-quality corpus to build a good quality translation system [1,2]. Therefore, NMT usually only works well for resource-rich languages, i.e., languages that contain several hundred thousand or millions of parallel sentences. Hindi, the language with the largest amount of parallel resources in India, is still considered a mid-resource language compared to its European counterparts. Other Indic languages, such as Bengali, have even fewer parallel resources and can be classified as *low-resource* languages. One popular technique that generates additional synthetic parallel data to build larger MT models even in the low-resource scenario is called back-translation. The authors of [3] leverage the ability of the encoder–decoder architecture to generate synthetic sentences by switching the direction of the translation model, i.e., translating the monolingual text in the target language to the source language (target-source), and reversing the sentence-pairs to create additional source-target data. This method relies on the quality of the target-language monolingual text and the architectures used to build the models. Considering the number of unknown factors, one promising research direction is experimenting with these variables and determining the efficacy of back-translation, given the amount of resources in Indic languages.

In this work, we build English-to-Indian language MT systems for Hindi and Bengali, using the state-of-the-art transformer architecture, and evaluate their performance. In addition, we incorporate the back-translation technique to analyse its impact on the system performance. The translation outputs are initially evaluated using automatic evaluation metrics, and then manually evaluated to analyse the efficiency of automatic evaluation.

The rest of this paper is organized as follows: Section 2 provides a detailed literature review. We formulate and discuss the research questions in Section 3. We discuss our system in Section 4. Section 5 provides all the details of our experiments. The results are highlighted in Section 6. We discuss output analysis in Section 7. Finally, we conclude our work and point out some future possible research avenues in Section 8.

## 2. Literature Review

### 2.1. Neural Machine Translation

The NMT architecture of Bahdanau et al. [1] is a sequence-to-sequence model that can be broken down into encoder and decoder blocks. The encoder network first generates word embeddings for each word in the source-language sentence. These word embeddings are numerical representations of the words in a multi-dimensional space. They allow for the model to encode the sequence of words in each sentence as distributed semantic representations. To regenerate the sequence in the target language, the decoder network regenerates the target-language sentence (token by token) from left to right. The encoder generates a fixed-length vector from an input sequence (X) in the source language, and a decoder uses it to decode and generate a translated sequence (Y) in the target language.

### 2.2. Transformers

The transformer architecture [2] uses the *Attention* mechanism to focus on the most important weighted words instead of every word in the input sequence. Their architecture entirely forgoes the Recurrent Neural Network's (RNN) ordinal memory, favouring the attention mechanism, which draws global dependencies between inputs and outputs. Before *Transformers*, RNN and *Long Short Term Memory* (LSTM) networks were the state-of-the-art in NMT. RNN was usually successful at modeling source sentence sequences to target sentence sequences, but was eventually replaced by LSTMS because the gradients of the network would explode and vanish if the length of the sequences increased beyond a particular threshold. The training was also hardware-intensive, even with a truncated back-propagation phase. Neural networks with LSTM cells performed well with longer sequences but were even harder to train because of their continued ingestion of serialised input. Naturally, they underutilised the parallelisation ability of GPUs.

The transformer architecture overcomes the drawbacks mentioned above by parallel ingestion of all the words in a sentence in a single time-step, unlike its predecessors. It follows the encoder–decoder architecture, where each layer in the encoder and decoder comprises a self-attention sub-layer, followed by a feed-forward network. The input and output embeddings are positionally encoded, i.e., the weight of each word in the sentence is calculated based on its distance from other words in the sentence, using any reasonable mathematical function. The feed-forward network is applied to all of the attention vectors. These feed-forward networks are used in practice to transform the attention vectors into a form that is digestible by the next encoder block or decoder block.

### 2.3. Alternative Low-Resource Solutions

Most data augmentation methods, such as back-translation, are specific to the data available in the language pair. Poncelas et al. [4] explore back-translated data as a separate standalone dataset, as well as combined with human-generated parallel data. They use incrementally larger amounts of back-translated data to train English-to-German NMT models and investigate the quality of the resulting translations. Fadaee et al. [5] use a dataset that only contains parallel bitext to augment data with a word replacement approach. It replaces words in the target sentences with rare words in the target vocabulary. Accordingly, the aligned source words are replaced in the source sentences based on the semantic rules of the language model. Other generic word replacement techniques include word dropout and Reward Augmented Maximum Likelihood (RAML) [6,7]. These methods share the common drawback of training brittle models resulting from noisy augmented data.

Alternatively, Zoph et al. [8] try to solve the low-resource problem by borrowing from a high-resource parent language. They train a high-resource language pair (the parent model), then transfer some of the learned parameters to the low-resource language pair (the child model) to initialize and constrain training. They further improve the results by using ensembling and unknown word replacement. Their results show that this technique only works well for highly related languages because most model components will have to be frozen or strongly regularized to be effective.

### 2.4. Related Work

A significant amount of effort has been put into the machine translation of low-resource Indian languages, but there has been little concentrated effort in Bengali. Choudhary et al. [9] modeled Tamil, Malayalam, Urdu, Bengali, and Telugu in their efforts to alleviate the low-resource problem. They use pre-trained Byte-Pair-Encoded (BPE) [10] and MultiBPE embeddings [11] to help solve the out-of-vocabulary (OOV) problem. Their method outperforms Google Translate by 60%. Goyal and Sharma [12] work with the Hindi language and use back-translation for the WAT-2019 Hindi–English shared task http://lotus.kuee.kyoto-u.ac.jp/WAT/WAT2019/index.html#task.html (accessed on 8 April 2022). Das et al. [13] assert that back-translation works better with similar languages. They use the LSTM Attention model [1] to build a Bengali-to-Hindi translation system. With a similar intuition, Przystupa and Abdul-Mageed [14] use back-translation on their Hindi–Nepali language pair and deterministically generate synthetic data using greedy decoding. Their results show that back-translation works better when the language pair has overlapping tokens, and the parallel bitext only contains short sentences.

### 2.5. Tools

In our experiments, (discussed later in Section 5), we use *OpenNMT* [15], an open-source NMT toolkit that supports a complete translation workflow, with features for training, language modeling, and decoding, including many others. The work of [16] adds the transformer model to their toolkit along with new configurations, such as copy attention and relative position. We use the default configurations of OpenNMT to build our translation models.

Several evaluation metrics exist for MT; however, each has its advantages and disadvantages. The Bilingual Evaluation Understudy (BLEU) evaluation metric [17] is the most widely used metric for MT evaluation. It is language-independent and closely mimics human evaluations. Unfortunately, they tend to favour shorter sentences by providing very high precision scores. The Metric for Evaluation of Translation with Explicit ORdering (METEOR) Universal [18] overcomes this bias using word-to-word matching between the target and the reference sentences. Additionally, it has stemming and synonymy matching features. The Translation Error Rate (TER) [19] is suited for post-editing tasks, because it assesses an MT output by calculating the number of changes a human translator would have to make in order for the MT output to match the reference sentence in meaning and fluency.

### 2.6. Languages

In this work, we conduct experiments on English and two Indian languages: Hindi and Bengali. These languages are chosen on the basis of diversity. They contain unique scripts and are spoken in different regions of India. There is also a significant difference in the amount of parallel English data available in these languages, which makes them close representatives of other languages with similar resources.

- **Hindi:** Hindi belongs to the Indo-Aryan language family and is a descendent of Sanskrit, like many Indian languages. Like Sanskrit, Hindi also uses the Devanagari script, although the script offers minimal phonetics to certain sounds. The sentence structure of short sentences in Hindi is flexible; in longer sentences, the Subject–Object–Verb structure is given preference.

- **Bengali:** Bengali is also a descendent of Sanskrit and from the Indo-Aryan language family, but it makes use of a custom script that is more phonetically suitable. It is not inflected by gender, has the same grammatical rules as Hindi, and is highly morphological. It is the most widely spoken Indian language after Hindi.

## 3. Research Questions

Our main goals in this work are to investigate the impact of back-translation in improving the English-to-Indian language MT system and also to evaluate the outputs using both automatic and human evaluation metrics. Therefore, we address the following two research questions (RQ).

- **RQ-1**: How efficient is back-translation in improving the baseline system built from the most recently developed largest known Indian language parallel corpus called *Samanantar* [20], especially for Hindi and Bengali?
- **RQ-2**: Is the actual translation quality similarly reflected in both *Automatic* and *Manual* evaluations?

To answer the first research question (RQ-1), we initially built a baseline system from *Samanantar* corpus. Afterwards, we built an extended model, which is trained from the concatenation of *Samanantar* and a synthetic parallel corpus that is produced by back-translating a monolingual corpus (details in Section 4). Finally, we evaluated both the baseline and the extended model in order to investigate if the the back-translation was capable of improving the performance of the baseline system. The second research question (RQ-1) was addressed by evaluating the translation outputs using both automatic and human evaluation and observing the scores.

## 4. System Description

### 4.1. Corpora Used

Many parallel corpora are available for MT development, but not all of them are of good quality. Although certain corpora show great promise in terms of the number of sentences and tokens, they train bad models due to the minimal variance among examples and limited complexity of sentence structure. After a careful corpus survey, we decided to use the *Samanantar* parallel corpus as the training data. For tuning and testing purposes, we used the latest benchmark 'WAT-2021' dataset http://lotus.kuee.kyoto-u.ac.jp/WAT/indic-multilingual/ (accessed on 8 April 2022) of English–Indian language MT evaluation.

To the best of our knowledge, *Samanantar* is the most extensive publicly available parallel corpora collection for 11 Indic languages. Table 1 shows the corpus statistics.

**Table 1.** Corpus statistics for training, development and test.

| Corpus Name | Number of Parallel Sentences per Language Pair | |
|---|---|---|
| | **English–Hindi** | **English–Bengali** |
| Samanantar | 8,466,307 | 8,435,355 |
| WAT-2021 development | 1000 | 1000 |
| WAT-2021 test | 2390 | 2390 |

For back-translation, we used 'IndicCorp' https://indicnlp.ai4bharat.org/corpora/ (accessed on 8 April 2022), one of the largest publicly available monolingual corpora for Indian languages.

### 4.2. Corpus Pre-Processing

Our training data were pre-processed using the following steps.

- **Filtering long sentences:** Extremely long sentences were deleted because MT systems generally produce a low-quality translation for very long sentences. If either side contains too many words (100 words is set as the default limit), the sentence pair is discarded.

- **Removing blank lines:** Sentence pairs with no content on either side are removed.
- **Removing sentence pairs with odd ratio:** Sentences with marginally longer or shorter translations compared to their original sentences were removed because of the probability of their being incorrect translations. The filtering ratio was 1:3 in our case.
- **Removing duplicates:** All duplicate sentence pairs were discarded.
- **Tokenisation:** We broke down the sentences into their most basic elements, which were called tokens. Tokenisation is particularly relevant because it is the form in which transformer models ingest sentences. In practice, most NMT models are fed with sub-words as tokens.
- **BPE:** Both the Indic languages we used in this study are derivatives of Sanskrit, which makes them morphologically rich. This would imply that most OOV words have similar morphemes to some of the words already in our vocabulary. With this in mind, the BPE technique was leveraged to resolve the OOV problem by helping the model infer the meaning of words through similarity. The BPE algorithm performs sub-word regularization by building a vocabulary using corpus statistics. Firstly, it learns the most frequently occurring sequences of characters, and then it greedily merges them to obtain new text segments.

## 5. Experiments

### 5.1. Building Baseline Models

The first stage of our experiments involves building baseline models in both directions (i.e., English to Indian language and vice versa) with the *Samanantar* corpus.

### 5.2. Building Back-Translation Models

All of our NMT models used the WAT-2021 development and test sets of Hindi and Bengali, respectively.

### 5.3. Parameter Settings for MT Models

As mentioned earlier, we used the *OpenNMT* tool to build the MT models in this work. We used the default parameter settings. Some of the parameter values are as follows:

- Minibatch size = 128;
- Hidden state size = 1000;
- Source and target vocabulary size = 32 K;
- Low dropout probability = 0.2;
- Learning rate used for both forward and backward models = 0.2;
- Decay rate = 0.9999;
- Beam search width = 12;
- Save checkpoint steps = 10,000;
- Minimum train steps = 100,000.

Although this a standard parameter setting, used in many cases, we varied some of the settings; for example, the minimum training steps were increased to 200,000. There was no improvement in the result. However, we did not explore the variations in all the parameters. It would be interesting to test different combinations of parameter values in order to investigate the system's performance in different settings.

### 5.4. Evaluation Metrics

Several automatic evaluation metrics are available, such as BLEU, METEOR, and TER. The BLEU metric calculates the score by comparing a candidate translation of text to one or more reference translations. A perfect match results in a score of 1.0, whereas a perfect mismatch results in a score of 0.0. METEOR is based on the harmonic mean of unigram precision and recall, with recall weighted higher than precision. It also performs stemming- and synonymy-matching, along with the standard exact word-matching. On the other hand, TER measures the number of actions required to edit a translated segment in line with one of the reference translations. The lower the TER score is, the better the translation

quality. In contrast, a higher BLEU or METEOR score represents better translation quality. In this work, we used BLEU as it is the most widely used automatic evaluation metric and is well correlated with human evaluation in most cases.

*5.5. Experimental Architecture*

Our experimental architecture using back-translation is illustrated in Figure 1. The whole system consists of two main parts; (i) translating monolingual data from Indian language (IL) to English (EN) to generate a synthetic dataset, and (ii) building an extended model from the combination of EN–IL parallel corpus and the synthetic dataset. The blue dotted lines with arrows show the process of generating synthetic parallel corpus by translating the monolingual corpora in Indian language (i.e., Monolingual IL) into English (i.e., Translated EN). This monolingual corpora is translated into English using the model built from the IL–EN parallel, which is shown in two circles connected with an arrow (i.e., IL->EN) right below the blue dotted lines. In the next part of this figure, we can see the original EN–IL parallel corpus in two bigger circles with an arrow.

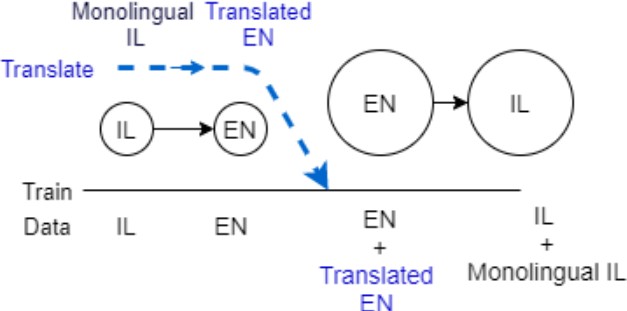

**Figure 1.** Experimental architecture.

Both of the source (EN) and target (IL) of this training data were then concatenated with their respective counterparts of the generated synthetic data set, i.e., the English texts (EN) were concatenated with the English translations of the monolingual IL corpus and the IL texts were concatenated with monolingual IL texts. Finally, an extended model was built by training this concatenated parallel corpus.

In summary, Figure 1 shows two stages of our experiments:

- **Stage 1:** This generates synthetic data by translating (back-translation) from Indian-language (IL) into English (EN), and
- **Stage 2:** This builds English-to-Indian language MT systems using the existing parallel data and the generated synthetic parallel data.

Our models' vocabularies were built using the sub-words generated after applying the BPE technique to our dataset. To generate synthetic data for our back-translation models, we first built Indian language to English translation models and then used them to translate sentences in our monolingual dataset. Afterwards, these synthetic data were added to the already existing parallel dataset. For example, we first built a Hindi-to-English translation model to translate Hindi monolingual sentences into English. The translations are aligned with the monolingual Hindi sentences to form the new synthetic dataset, which was then concatenated with the original English–Hindi parallel corpus to form the extended corpus. Our extended models were then trained from scratch with new vocabularies built using the extended dataset. The translation outputs were evaluated using BLEU. The higher the BLEU score, the better the translation quality.

**6. Results**

*6.1. Automatic Evaluation*

The results of automatic evaluation using the BLEU score in our experiments are shown in Table 2.

**Table 2.** BLEU scores for English-to-Indian language translation models on Samanantar dataset.

| Translation Model | BLEU Score per Language Pair | |
|---|---|---|
| | English–Hindi | English–Bengali |
| English – Indian language | 33.45 | 11.58 |
| English – Indian Language + back-translation | **33.72** | **11.99** |

The best BLEU scores for English-to-Indian language outputs are shown in bold font. The scores show that models which already have strong baselines, such as the English-to-Hindi model, improve, if only by 0.27 (0.8% relative improvement), upon the application of back-translation. We hypothesise that this improvement could be attributed to the strength of the Hindi-to-English translation model built from the *Samanantar* corpus and the high-quality translations of the monolingual data that it produces. A slightly more BLEU score increment (0.41, 3.5% relative improvement) was noticed for the English-to-Bengali model using the back-translation technique. This indicates that back-translation is more helpful to improve an already weaker MT model (low-scoring model, i.e, English-to-Bengali in this case) as compared to the already high-scoring model, i.e, English-to-Hindi in this case.

In general, the models translating sentences into English performed better. A probable reason for this is that English is relatively morphologically impoverished when compared to many Indian languages. It is, therefore, easier to translate from Indian languages to English than in the opposite direction. The BLEU scores are shown in Table 3. The authors of *Samanantar* noticed a similar phenomenon and attributed this to the improved transference in many-to-one settings compared to one-to-many settings [20].

**Table 3.** BLEU scores for Indian-to-English language translation models on the Samanantar dataset.

| Translation Model | BLEU Score |
|---|---|
| Hindi-to-English | 38.57 |
| Bengali-to-English | 22.84 |

*6.2. Manual Evaluation*

While automatic evaluation metrics such as BLEU are convenient and easy to use, they cannot always capture the semantic similarity between the translation output and the reference, due to the complexity of meanings. This is amplified when the source and the target language are significantly different in terms of structure and meaning representation. As we deal with such languages in our work, it is very important to manually evaluate the translation quality to validate the BLEU scores. However, manual evaluation is a significantly time-consuming task when multiple languages are involved. Bearing this in mind, we manually inspected a subset of 200 translation outputs generated by each of the MT systems for both English–Hindi and English–Bengali language pairs in both translation directions resulting in a total of 800 evaluation outputs (i.e, outputs from 4 MT models). We evaluated them using the 'Adequacy' and 'Fluency' metrics, explained in brief as follows.

- **Adequacy**: This refers to the measurement of how much information is retained in the translation outputs as compared to the references, regardless of the grammatical correctness.
- **Fluency**: This refers to the measurement of how fluent the output is, that is, how grammatically correct it is, regardless of adequacy.

We used a 5-point scale for both adequacy and fluency. Table 4 shows a description of the scale for each metric. According to our 5-point scaling system, any translation output that is assigned both adequacy and fluency scores of more than 3 is considered to be a good-quality translation. In contrast, all the other translations with equal to or less then 3 points for both adequacy and fluency can be considered average and poor, respectively. It is obvious that an ideal translation must have both adequacy and fluency scores of 5 points.

**Table 4.** Description of Adequacy and Fluency.

| Scale | Adequacy | Fluency |
|---|---|---|
| 5 | all information present in the translation | perfect in terms of grammatical correctness |
| 4 | most of the information present | not perfect but very good |
| 3 | nearly half of the information present | average quality |
| 2 | very little information present | poor quality |
| 1 | no information present | worst or completely incomprehensible |

It is important to note that a good translation should convey all this information while being fluent. Fluency is meaningless without adequacy. For this reason, we do not consider a translation output that has high fluency and low adequacy and vice versa and as an 'unconsidered' translation. Among 200 manually evaluated translations, we found only one to be an 'unconsidered' translation Note that this single 'unconsidered translation' was observed in 200 outputs; the number may increase if a bigger subset is evaluated, which is shown in Table 5.

**Table 5.** An example of unconsidered translation for Hindi-to-English.

| Reference | MT Output | Adequacy | Fluency |
|---|---|---|---|
| "In a way, this is endowed with the might to transform the entire season-cycle of the country". | It is the weather of the entire country. | 2 | 5 |

We can see in the above table that only a little information is retained in the output; hence, the adequacy is 2 according to our evaluation criteria. In contrast, the output is completely fluent (regardless of adequacy) and so was assigned a fluency score of 5. As the output eventually does not make any sense, even when completely fluent, it is treated as an 'unconsidered' translation.

The average adequacy and fluency score for the English–Hindi and English–Bengali pairs in both directions are shown in Table 6. We notice that the Hindi-to-English translation model produces the best adequacy and fluency scores of 4.33 and an average fluency score of 4.83. This is also true for BLEU scores for Hindi-to-English outputs, which were the highest of the two scores (see Table 3).

**Table 6.** Results of manual evaluation.

| Language Pair | Translation Direction | Average Adequacy | Average Fluency |
|---|---|---|---|
| **English–Hindi** | English-to-Hindi | 4.31 | 4.77 |
| | Hindi-to-English | **4.33** | **4.83** |
| **English–Bengali** | English-to-Bengali | 3.74 | 4.4 |
| | Bengali-to-English | 3.87 | 4.74 |

Now, if we compare the results of Table 6 with those of Table 2, we notice that the BLEU score for English-to-Bengali is 64% less than that of English-to-Hindi, but the average adequacy is 13% and average fluency is nearly 8% less, respectively. Similar observations can be made when the manual evaluation results of Hindi-to-English and Bengali-to-English are compared with the BLEU scores of Table 3. These results show inconsistencies between automatic and manual evaluations, and hence prove that BLEU scores are not always reliable for MT evaluation.

As mentioned earlier, the translations are only considered *good* when both adequacy and fluency are greater than 3, i.e, when they retain all or most of the information and are also fluent or nearly fluent. All other translations are considered either *Average* or *poor*; the

translations that are unable to retain most of the information are less fluent. Table 7 reflects the overall translation quality considering both of these metrics.

**Table 7.** Quality of translation outputs.

| Language Pairs | Translation Direction | Translation Quality Good | Average or Poor |
|---|---|---|---|
| English–Hindi | English-to-Hindi | 90.5% | 9.5% |
| | Hindi-to-English | 92.5% | 7.5% |
| English–Bengali | English-to-Bengali | 68% | 32% |
| | Bengali-to-English | 78% | 22% |

It can be seen that 90.5% of translations produced by English-to-Hindi model are of good quality. The amount of average or poor translations is much lower, below 10%. Slightly better results were noticed for the Hindi-to-English model. It is obvious that both of these models produce a high percentage of good-quality translations. On the other hand, the scores for English-to-Bengali (68%) and Bengali-to-Hindi (78%) are significantly less than those for Hindi, but are still good enough, as most of the outputs are good quality.

## 7. Output Analysis

For the sake of reading simplicity, we now show some example outputs of Indian_ Language-to-English instead of English-to-Indian language MT systems in Tables 8 and 9.

**Table 8.** Some example Hindi-to-English translation outputs with adequacy and fluency scores.

| Examples | Reference | MT Output | Adequacy | Fluency |
|---|---|---|---|---|
| 1 | I recently visited the Krishi Unnati Mela organized in New Delhi. | I had visited the Agri-Unnati Mela in Delhi recently. | 4 | 5 |
| 2 | Start-Ups have been given income tax exemption for three years. | Startups are exempted from paying income tax for 3 years. | 5 | 5 |
| 3 | Yoga helps to maintain balance amidst this disintegration. | Adds yoga between this scatter. | 2 | 2 |
| 4 | "It brings about peace in the family by uniting the person with the family." | It brings happiness and prosperity to the family. | 3 | 5 |

In the first example of Table 8, we can see that the MT system produces a slightly incorrect translation "Agri-Unnati Mela" as compared to the reference. However, it retains most of the information and so is assigned an adequacy score of 4 and a fluency score of 5 as a fluent translation. The second example is a perfect translation because it retains all the information and is completely fluent; thus, it achieves a score of 5 for both fluency and adequacy. The third example obtains very low scores as it loses most of the information and is not fluent. The final example is fluent but retains only nearly half of the information, and so cannot be considered a good translation.

**Table 9.** Some example Bengali-to-English translation outputs with adequacy and fluency scores.

| Examples | Reference | MT Output | Adequacy | Fluency |
|---|---|---|---|---|
| 1 | "Not only this, we are also the sixth largest producer of renewable energy". | We have also earned the honour of the sixth largest producer of renewable energy. | 4 | 5 |
| 2 | Start-Ups have been given income tax exemption for three years. | Start-up companies have been given tax concessions for the first three years. | 5 | 5 |
| 3 | "In a way, this is endowed with the might to transform the entire season-cycle of the country. | All these ideas are the strength of the country's transformation. | 1 | 3 |
| 4 | "That is why a large number of letters on agriculture have been received". | Many letters have been written about agriculture. | 3 | 5 |

In Table 9, we can see that the first and the second examples are good-quality translation outputs, and achieve high adequacy and fluency scores. In contrast, the third and the fourth examples fail to fulfil the criteria of being good-quality outputs.

## 8. Conclusions and Future Work

In this work, we built English-to-Indian languages MT systems using the state-of-the-art transformer architecure and subword NMT. We applied back-translation with the aim of improving the performance of our MT systems. We reported positive improvements in the BLEU scores with back-translation. It was observed that back-translation technique helps the weaker MT models more than it helps already strong models. MT models with English as the target language performed better than those in the opposite direction. This produces good-quality synthetic data by translating the monolingual corpus in Indian languages into English, and helps to improve the quality of baseline English-to-Hindi and English-to-Bengali translation systems using back-translation. In addition, we performed a manual evaluation of a subset of 200 translation outputs for each translation model in order to test the efficiency of the BLEU metric. We observed that BLEU does not always correlate well with human evaluation. BLEU scores did not reflect the actual quality of English-to-Bengali translation. The outputs were, in fact, better than those evaluated by BLEU. Our work can be extended by exploring monolingual datasets of different sizes and domains to precisely identify the saturation point of back-translation. We also plan to perform a manual evaluation on a bigger set of translation outputs. In addition, we will extend our comparison to not only the baseline but to other state of the art methods.

**Author Contributions:** Methodology, A.K. and P.L.; Resources, S.K.M.; Supervision, A.W. All authors have read and agreed to the published version of the manuscript.

**Funding:** This research was funded by Science Foundation Ireland grant number 13/RC/2106.

**Institutional Review Board Statement:** Not applicable.

**Informed Consent Statement:** Not applicable.

**Data Availability Statement:** Not applicable.

**Conflicts of Interest:** The authors declare no conflict of interest.

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
