# Peer review of "Improving English-to-Indian Language Neural Machine Translation Systems"

_information, doi:10.3390/info13050245_

Round 1

Reviewer 1 Report

The authors aim to build machine translation systems using the new transformer machine learning model for Hindi and Bengali,which are not resource-rich languages, and assess their performance. It also investigates the effects of back-translation on these systems and compares automatic and manual evaluation of the quality of translation.

In my view, these research questions could have been listed and then answered more clearly - in one section. The system is described quite well. Figure 1 should have ben described in more detail, though. It seems to me that it is quite difficult to quantify the quality of translation, although the authors explain their approach clearly.

All in all, this research is trendy, we have seen massive improvements in machine translation of late.  In case of low-resource languages it is vital to look for ways how to improve machine translation, which is exactly what the authors want to do.  

As for the English language, I have found only a few slips: e.g., analyse as good as (line 8), is very less (line 276).   

Author Response

1) We will list the research questions and then answer them more clearly in the next version

2) We will described Figure 1 in more detail

3) We will fix the typos

Reviewer 2 Report

Presented paper describes the solution for improvement of English to Indian language translation using neural machine translation systems.

In general, the paper is rather easy to follow and written in comprehensible fashion, however, I have several remarks regarding to the paper content. First, the overal description of the used approach is very shallow. Authors do not properly describe the models used, their parameters etc., instead they provide just overall description of the given approach (mostly in section 4). Also, many used terms (either models, or metrics) are not explained properly in the manuscript, just briefly introduced and referenced (e.g., BLEU score and others). Related work focuses on translating of Indian laguages - however, it could be beneficial to demonstrate the overall area, including other approaches, possibly applied on different (but maybe similar) translations. Finally, the paper title says "Improving ...", therefore I would expect to actually compare the results of presented experimetns with the other methods (not just the baseline model), to show the actual improvement over the other state of the art methods.

Multiple formal remarks:

  • some tables/figures aligned left, some to center
  • structure of the paper is not coherent (usage of the styles for paragraphs, etc.)
  • some abbreviations not explained 

Author Response

1) We will describe the models and the parameters used in details in the next version

2) We will explain BLEU mertics and others in details and also provide further explanations on scores

3) The tables and figures will be formatted properly.

4) The abbreviations will be explained in details.

Reviewer 3 Report

The work present is a re-submission of a previous manuscript.

This time it presents enough information to be marginally accepted, the overall innovation is still low, it is a practical evaluation and tunning of a transformer topology to the specific task of translation between Indian and English languages. Regardless it can be interesting for some researchers.

Author Response

Yes, its the resubmission of the previous manuscript. However, there are many changes in the content, specially the results. We performed experiments that are different from the previous ones.

Thanks for the suggestions, we have highlighted the further possible improvements as future work.

Round 2

Reviewer 1 Report

Corcerning the mentioned "spelling check required" I mean the newly added text, e.g., lines 153 an unnecessary space (corpus . ...) or 204 "we" instead of  "We" Otherwise, I think the revised text is fine.

Reviewer 3 Report

My main concerns were addressed.

This manuscript is a resubmission of an earlier submission. The following is a list of the peer review reports and author responses from that submission.

Round 1

Reviewer 1 Report

the authors built several English-Indian languages Machine Translation systems using the state-of-the-art transformer architecture, and conclude that the state-of-the-art NMT architecture helps produce good quality fluent translation outputs even for low-resource Indic languages.

the paper investigated the use of back-translation as a means to improve quality of MT between low-resource languages, and found that Hindi was the only Indic language which showed an improvement of 0.03 in terms of BLEU, which is insignificant. So from the experiments the back-translation is almost useless, which contradicts with most work in the field. 

from the authors' findings we can conclude that BLEU scores are inadequate to represent the translation quality, since large gaps in BLEU score do not indicate an equally large gap in translation quality as measured by manual evaluation. This is again deviating from major literatures. But the paper does not tell us on which dataset the manual evalatution is performed, since the BLEU gaps between the PIB and Samanantar are very large for Bengali and Telugu. 

Although the corpus sizes are almost same for Hindi and Bengali (about 8 millions sentences), doubling the size of Telugu, in the Samanantar corpus, there are large gaps for the EN-IL BLEU scores, i.e. 29.83, 10.21, 15.84 respectively for Hindi, Bengali and Telugu. The authors failed to analyze the big differences here. Same corpus size, big BLEU score gaps: 29.83 v.s. 10.21; smaller corpus size while higher BLEU scores: 10.21 v.s. 15.84. The scores make the 0.03 improvement earlier almost meaningless.

For back-translation, more expriments are needed. Table 4 only lists results for the En–IL direction, the reverse direction is also needed. The paper says in the conclusion section that we observed that target-to-source translation models need to be significantly stronger than the source-to-target translation models to yield a positive result. Without back-translation in both directions, this concluson is lacking support.

the authors mentioned the domain mismatch between the monolingual data used for back-translation and bilingual data, but does not provide any experiments. This is another import aspect that needs to be addressed.

My suggestion is that more experiments are needed to clarify the above questions.

Author Response

Thank you for your review and your insights. I will try to address each concern to the best of my ability.

Concern 1- We state that we saw an improvement of 0.03% when we applied back translation on our English to Hindi  model which was trained on the Samanantar data set. However you suggest that it would be better if we refrain from stating this because the improvement is insignificant.

Repy 1- We agree and we will rephrase it.

Concern 2-  The paper does not tell you on which dataset the manual evalatution is performed, since the BLEU gaps between the PIB and Samanantar are very large for Bengali and Telugu.

Reply 2- Manual evaluation was done on the output from the best models of each language. Because these happened to come from Samanantar trained models, these are the only models we evaluated.

Concern 3-   the BLEU gaps between the PIB and Samanantar are very large for Bengali and Telugu.

Reply 3- I believe the huge gaps in  PIB and Samanantar occurred because of the size of the datasets.

 Concern 4- The authors failed to analyse the big differences in the BLEU scores of the best models (29.83 ,10.21, and 15.84) despite of having the same number of sentences.  

Reply 4- We chose to do a manual evaluation of these models precisely to understand these differences. After a manual evaluation we found that there were two reasons for this. We found that in certain cases, because of the free word order in Indian languages most sentences were penalised by the BLEU evaluation metric despite being correct translations. This is because the original sentence in the Indian language used a different word order than the sentence produced by the model. I am assuming that the reader of this response may not read Hindi/English/Telugu. I am therefore giving you an example using the outputs generated by the Hindi/Telugu/Bengali to English model.

If you can see the example highlighted in yellow, you will see how the BLEU evaluation metric heavily penalises sentences with a different word order. We noticed the same phenomenon occurring,  very often, in the Indian language translations. It is why we believe that the sentences generated by the Telugu and Bengali model were better than the BELU score indicates. To prove this we would need alternative translations of the test sentences that use a different word order, unfortunately they are not currently available in most Indian languages.

Translation
System
Outputs Reference Adequacy Fluency BLEU
Telugu-to-English The President of Meghalaya is the
President of India.
On the special occasion of their Statehood Day
, greetings to the people of Meghalaya
2 3 6.16
The situation is being monitored regularly The situation is being monitored closely 4 5 75.98
Hindi-to-English The situation is being closely monitored The situation is being monitored closely 5 5 56.23
This app is also beneficial for businesses The App is also beneficial for traders 4 5 61.48
Bengali-to_English We are proud of the World Environment
Day 2018
We are proud to be the global host for World
Environment Day, 2018
3 4 18
It will provide quality healthcare to the poor. It will provide top quality healthcare to the poor, he added. 4 5 37.01

Concern 4- Same corpus size, big BLEU score gaps: 29.83 v.s. 10.21; smaller corpus size while higher BLEU scores: 10.21 v.s. 15.84.

Reply 4 - Unfortunately, we cannot identify precisely what causes this scenario, and hypothesise that the the nature of the data caused it. The Samanantar paper also has a similar trend, where, Telugu despite having fewer sentences performs better than Bengali.  The presence of this pattern in both the results lead us to believe that perhaps the Bengali parallel corpus is a poor dataset despite having several sentences. the following are the results that xxxx got when they trained their model Indictrans on samanantar. Their scores are much higher than ours because indictrans was trained using different configurations. 

language BLEU score
Hindi 43.2
Bengali 28.4
Telugu 35.1

Concern 5- We mentioned the domain mismatch between the monolingual data used for back-translation and bilingual data, but does not provide any experiments. 

Reply 5- We mentioned that there was a Domain mismatch- because during our literature review, we read that the IndicCorp data is India centric. We therefore came to the conclusion that perhaps the sentences had a lot of unseen vocabulary that the model had not seen earlier. We now know that this was a rather broad assumption and we would like to remove this sentence from our paper.

Concern 6- My suggestion is that more experiments are needed to clarify the above questions.

Reply- Unfortunately we are not in a position to do new experiments because, the resources allocated to us are no longer available.

I hope this response will help you understand our choices better this time round. We look forward to your response.

Thanks and best regards,

Akshara

Reviewer 2 Report

This article proposal is innovative as it aims to investigate the utility of building English-Indian Languages Machine Translation Systems and evaluate backtranslation.

However, I believe the paper needs to be amended due to a serious of factors before performing a more in-depth review: 

  • Although the need of including Hindi, Bengali and Telugu was made clear in the research (particularly Bengali and Telugu as ‘low-resource’ languages), the aim may seem too difficult to achieve in a single paper. Hindi, described as a ‘mid-resource language’, belongs to the Indo-aryan languages subgroup of IE language family whereas Bengali and Telugu are two Dravidian languages, South-central in the case of Telugu and Eastern in the case of Bengali (Eastern Magadhan). Therefore, there is a need to clarify different concepts and variables used in the paper given the linguistic distance, particularly since the authors citing Zoph et al. (2016) mention that some techniques employed in Neural Machine Translation (NMT) work ‘well only for highly related languages …’ (117-118).

  • The authors should clarify the possible impact and limitations on the results of the corpora they used given the differences between the parallel sentences included in each case, particularly the English-Telugu (Samanantar corpus, 4,775,516) as opposed to English-Hindi (8,466,307) and English-Bengali (8,435,355). Similarly, they should further explain the difference in the corpus size (short sentences) and its impact between Hindi (590,892) as opposed to Telugu (1,644,651) and Bengali (1,403,421) in Indicorp used for backtranslation (Table 2). Please, consider the following statement ‘Indian languages have a free word order and therefore when only provided with one reference for evaluation, it us (SIC) likely that the BLEU metric discredited good translations.’

  • In line with the previous comment, the authors should clearly explain if they detected any significant difference in corpus pre-processing between the 3 languages, particularly as regards removing sentence pairs with odd ratio.

  • The authors should first explain the domains of the corpora they selected and the relatedness in the three languages because it may seem to have a significant impact on the results as later acknowledged by them ‘Our second hypothesis is that back-translation did not improve any PIB-trained models because the back-translated data was not from the news domain, of poor quality (especially in Telugu and Bengali), and applied in a very high ratio’ (274-276).

  • Given the fact that the authors are using three languages belonging to different language families, there is clear need to explain the manual evaluation (process, human component, etc.) carried out for the ‘adequacy and fluency’ metrics. The authors may have opted for the adequacy-fluency metric instead of other more reliable but time-consuming methods such as HTER (Human-mediated Translation Edit Rate), which can be more helpful when combined with BLEU and METEOR, so they need to provide further details and clarify in this case the procedure and impact that human annotators may have on their research results.

Minor revision required, see some examples below:

- ‘The layers in the encoders use self-attention, which tries to answer is how relevant the ith word in the English sentence is relevant’ (83-84) – ‘IS’

- ‘it us likely that the BLEU metric discredited good translations’ (282) – ‘US’

- ‘These outputs are categorise them as ‘inconsiderable’ translations.’ (311-312) ‘CATEGORISE’

Author Response

Hello, I hope the following responses answer your concerns 

concern 1 - There is a need to clarify different concepts and variables used in the paper given the linguistic distance, particularly since the authors citing Zoph et al. (2016) mention that some techniques employed in Neural Machine Translation (NMT) work ‘well only for highly related languages …’ (117-118). 

Reply 1- We probable weren't coherent because we failed to mention that  the languages need to be when one is using transfer learning, like Zoph et al (2016), in their cross-lingual transfer learning paper. We put that in the paper just to highlight other low resource alternatives. Backtranslation does not need the language pairs to be related to  the best of our knowledge.

Concern 2- The authors should clarify the possible impact and limitations on the results of the corpora they used given the differences between the parallel sentences included in each case.

Reply 2- We started out with the impression that all the datasets were of almost similar quality sparing the number of sentences in each language pair. However after we saw that the Telugu dataset performed much better with fewer sentences, we now know that high quality sentences are as important as having a large dataset.

Concern 3-Similarly, they should further explain the difference in the corpus size (short sentences) and its impact between Hindi (590,892) as opposed to Telugu (1,644,651) and Bengali (1,403,421) in Indicorp used for backtranslation (Table 2)

Reply 3- Given that we had identified our best models with each language pair. we wanted to answer a few questions. the following are the questions we intended to answer with -

1) Applying a small amount of synthetic data to a strong model trained with high quality data (It is why we chose the Hindi-English Samanantar model and applied comparatively few sentences).

2) Applying a large amount of synthetic data to a weak model trained with low quality data (It is why we chose the Bengali-English Samanantar model and applied comparatively more sentences).

3) Applying a large amount of synthetic data to a strong model trained with high quality data, but fewer sentences, (It is why we chose the Telugu-English Samanantar model, and applied comparatively few sentences).

Concern 4- Please, consider the following statement ‘Indian languages have a free word order and therefore when only provided with one reference for evaluation, it us (SIC) likely that the BLEU metric discredited good translations.’

Reply 4- Thank you for pointing out the typing mistake.

Concern 5- The authors should clearly explain if they detected any significant difference in corpus pre-processing between the 3 languages, particularly as regards removing sentence pairs with odd ratio.

Reply 5- We did not use any language specific pre-processing and applied the same script for every language. We now see that this could have improved our paper further. We deleted sentences where the ratio of the number of words in each sentence was greater than 1:4 because we felt that the probability of them having the same information is quiet low.

Concern 6-The authors should first explain the domains of the corpora they selected and the relatedness in the three languages because it may seem to have a significant impact on the results as later acknowledged by them ‘Our second hypothesis is that back-translation did not improve any PIB-trained models because the back-translated data was not from the news domain, of poor quality (especially in Telugu and Bengali), and applied in a very high ratio’ (274-276)

Reply 6- We would like to retract what we said about the the PIB dataset being from the NEWS domain. We wanted to say that the language used in that dataset was of very high quality and very formal.

Concern 7- There is clear need to explain the manual evaluation (process, human component, etc.) carried out for the ‘adequacy and fluency’ metrics. The authors may have opted for the adequacy-fluency metric instead of other more reliable but time-consuming methods such as HTER (Human-mediated Translation Edit Rate), which can be more helpful when combined with BLEU and METEOR, so they need to provide further details and clarify in this case the procedure and impact that human annotators may have on their research results.

Reply 7-We did a manual evaluation just to verify if the translations were indeed as bad as the BLEU scores indicated.  We see the merit of evaluating our work using HTER instead. I believe that it is impossible for us to do it now and therefore we would like to mention this in the future work section of our paper.

Thankyou for your time. I look forward to heading back from you.

Best reagrds,

Akshara.